

# The technique of fuzzy analytic hierarchy process (FAHP) based on the triangular q-rung fuzzy numbers (TR-q-ROFNS) with applications in best African coffee brand selection

Yupei Huang[1], Muhammad Gulistan[2], Amir Rafique[3], Wathek Chammam[4], Khursheed Aurangzeb[5] and Ateeq Ur Rehman[6]

[1] College of Economics and Management, Zhejiang Normal University, Zhejiang, China
[2] Department of Electrical and Computer Engineering, University of Alberta, Edmonton, AB, Canada
[3] Department of Management Sciences, COMSATS University Islamabad, Islamabad, Pakistan
[4] Department of Mathematics, College of Science Al-Zulfi, Majmaah University, Majmaah, Saudi Arabia
[5] Department of Computer Engineering, College of Computer and Information Sciences, King Saud University, Riyadh, Saudi Arabia
[6] Department of Mathematics and Statistics, Hazara University, Mansehra, Pakistan

Corresponding author
Muhammad Gulistan,
mgulista@ualberta.ca

## ABSTRACT

The African coffee market offers a rich and diverse range of coffee profiles. The coffee producers of Africa face numerous challenges like climate change, market fluctuations, diseases, soil degradation and limited access to finance. These challenges badly affect their productivity, quality and livelihood. There are different factors like social and cultural, which can affect the coffee production. This study aims to develop multi criteria decision making (MCDM) methods and their applications in coffee market specifically in identifying factors influencing consumers' coffee brand preferences in South Africa, which is known for its vibrant coffee culture. For this purpose, first we developed the technique of analytic hierarchy process (AHP) in the environment of triangular q-rung orthopair fuzzy numbers. The triangular q-rung fuzzy numbers can effectively handle the uncertainty. The AHP technique has widely been used in decision making due to its flexibility in assigning weights and dealing with vagueness. The weights of critera plays a very important role in an MCDM problem. The development of AHP technique in triangular q-rung orthopair fuzzy environment can improve the decision making (DM) by handling vagueness in data and by using the most appropriate weights. Furthermore this new proposed method improves accuracy and minimize the information loss. The newly peoposed method is applied to different MCDM problems and comparative analysis is conducted to check the validity of results.

# INTRODUCTION

Consumer buying behavior is a multifaceted field of study that remains challenging to fully comprehend, given its close ties to the complexities of human psychology (*Kardes, Cline & Cronley, 2011*; *Kotler & Armstrong, 2009*). While forecasting consumer buying behavior is feasible through analyzing prior purchase data, uncovering the fundamental drivers behind these decisions remains a pursuit (*Solomon, 2004*). Decision-makers consider many psychological, personal, social, and cultural dimensions (*Solomon, 2004*). The recent surge in health and well-being awareness, particularly considering the COVID-19 pandemic, has introduced a new dimension to consumer preferences, including their expectations of health-related benefits from products like coffee. The global coffee market, estimated at $460 billion in 2022, is projected to reach an impressive $537.1 billion by 2025, marking a compound annual growth rate (CAGR) of approximately 5.3% during this period. South Africa, known for its vibrant coffee culture, has experienced a noteworthy uptick in coffee consumption in recent years. The South African Coffee Industry Landscape Report for 2022 offers a comprehensive overview of the coffee industry, encompassing the entire value chain, market size, forecasts, trends, innovations, key players, distribution networks, pricing dynamics, and more. In 2021 the coffee market's retail value grew compared to the previous year, with an anticipated average growth rate of 7.9% from 2022 to 2026. Several factors contributed to this growth, including heightened health consciousness during the pandemic and increased competition among coffee producers. Within this burgeoning market, various coffee brands have garnered significant demand, such as Assagay Coffee, Douwe Egberts, Jacobs Kronung, Colombo Coffee, and Frisco. Different stakeholders from coffee producers to importers and exporters interact and influence each other. For instance, consumer demand affects producers decision while government policies has implications on entire value chain. This study centers on the decision-making process concerning selecting these coffee brands and the attributes associated with each. When individuals decide among available alternatives in coffee selection, they weigh numerous factors, making it intriguing to understand the factors influencing their choices. Consequently, this study's primary objective is to identify the determinants of coffee brand selection decisions in South Africa. Furthermore, this study seeks to uncover the substantial impact of cultural, social, personal, and psychological factors on coffee selection decisions in South Africa. It also aspires to pinpoint the best coffee brand in South Africa and across different countries, considering diverse social and cultural values. The aim is to develop a practical approach or method for ranking various coffee brands, facilitating enhanced decision-making. Consumer decision-making involves evaluating several factors influencing individuals and groups when selecting different products (*Solomon, 2004*). It is essential to discern what consumers buy and comprehend how the decision-making process unfolds and how it affects their behavior (*Solomon, 2004*). Among the factors examined in this study, we focus on four key elements influencing coffee brand selection: product quality, convenience, brewing method, and price. Additionally, we consider social, cultural, personal, and psychological factors that can sway brand selection decisions. These factors encompass family preferences, reference points, social status, age, income levels, lifestyle choices, personality traits, occupation, cultural influences, subcultural affiliations, and social class distinctions. Notably, social

factors play a substantial role in shaping purchasing decisions (*Perreau, 2014*). Every individual is a part of society; as such, they are influenced by various social factors. These encompass family dynamics, reference groups, social roles, and social status, profoundly affecting buying behavior (*Perreau, 2014*). Social factors also play a pivotal role in shaping attitudes and opinions related to decision-making. Cultural factors, including overarching cultural norms, subcultural affiliations, and social class distinctions, influence consumer choices significantly. Moreover, personal, and psychological factors, such as age, income levels, lifestyle choices, personality traits, and occupation, mold individual decisions, including coffee brand preferences. The consumer decision-making process typically encompasses five stages: need recognition, information gathering, evaluation of alternatives, purchase decisions, and post-purchase evaluation (*Solomon, 2004*). Social, cultural, personal, and psychological factors influence each stage, underscoring the importance of rigorously studying these factors. This study undertakes this task, employing advanced methodologies to rank different coffee brands based on these multifaceted factors. Our approach integrates the q-rung fuzzy analytic hierarchy process (AHP) method (*Saaty, 1990b*) to identify the optimal coffee brand, leveraging an array of criteria and sub-criteria. In his seminal work in 1980, *Saaty (1990b)* introduced the AHP, which has since been instrumental in solving intricate decision-making processes. AHP has found applications in selecting the best alternatives and prioritizing objectives in operational research (*Chang, Wu & Chen, 2008*). This versatile technique enables decision-makers to make informed choices based on attribute values (*Saaty, 1990a*). It simplifies complex problems by structuring them into hierarchical frameworks comprising objectives, criteria, and sub-criteria, providing a fundamental scale for comparing magnitudes. AHP facilitates quantitative and qualitative measurements, making it suitable for addressing complex problems by breaking them down into hierarchical structures. This method ranks alternatives, offering sound decision-making procedures, well-defined criteria, and sub-criteria. *Liang (2003)* extended AHP in 2003, introducing it as a multi-attribute decision tool that accommodates quantitative and qualitative measures. A new fuzzy-AHP-based technique was proposed and a hierarchical system structure was constructed for the Massively Multiplayer Online Role-Playing Game design (*Lo & Wen, 2010*). In situations with ill-defined problems, crisp sets give way to a fuzzy approach. In a multi-attribute decision-making (MADM) context, AHP offers a mathematical foundation for judgment and reasoning, leveraging a ratio scale (*Ju, Wang & Liu, 2012*). The AHP adheres to three primary principles: decomposition, relative assessment, and fusion of priorities (*Ju, Wang & Liu, 2012*). To simplify decision-making problems, decision-makers often employ Saaty's nine-point scale for criterion selection. In practical applications, decision-makers may require assistance determining crisp estimate values for comparison purposes. The fuzzy analytic hierarchy process (FAHP) was introduced by *van Laarhoven & Pedrycz (1983)*, where satisfaction levels were represented in triangular numbers. They employed a logarithmic least squares approach to derive fuzzy weights and scores, aiding in evaluating alternatives. This approach effectively manages indeterminacy, inconsistency, and inaccuracy. Subsequently, an expanded traditional FAHP by introducing trapezoidal fuzzy numbers and calculating fuzzy weights and score functions using the geometric mean technique (*Jaiswal et al., 2015*). *Chang (1996)* extended this

 

methodology by using row averages of priorities, employing triangular fuzzy numbers. FAHP has been adopted across various fields, including administration (*Saaty, 1998*), the airline industry (*Rezaei, Fahim & Tavasszy, 2014*), lubricant regenerative technology selection (*Hsu, Lee & Kreng, 2010*), textiles (*Cebeci, 2009*), computer-aided machine-tool selection (*Duran & Aguilo, 2008*), silicon wafer slicing (*Chang, Wu & Chen, 2008*), transportation (*Kulak & Kahraman, 2005*), and numerous others (*Liu, Wang & Liu, 2018*; *Teng, Liu & Liu, 2018*; *Liu et al., 2021*). A new approach to handling fuzzy AHP was introduced using triangular fuzzy numbers and an extent analysis method (*Chang, 1996*). Multi-attribute decision-making (MADM) is the procedure for ranking finite alternatives based on different alternative values (*Liu & Wang, 2018*). Decision-makers often require support in obtaining precise alternative values. In response, *Zadeh (1965)* introduced fuzzy sets (FS), which have garnered attention from researchers. Although FS theory primarily deals with satisfactory grades, non-satisfactory grades are often needed. To address this, researchers introduced various extensions, such as intuitionistic fuzzy sets (IFS), Pythagorean fuzzy sets (PFS), and q-rung orthopair fuzzy sets (q-ROFSs), designed to handle incomplete and uncertain data. Atanassov pioneered IFS based on FS, wherein the sum of satisfactory and non-satisfactory grades falls between 0 and 1. This framework effectively addresses uncertain and vague data in decision-making scenarios. A comprehensive exploration of IFS included:

- Fundamental theories.
- Operational laws of IFS (*Biswas, De & Roy, 2000*).
- Distance measures between IFS (*Chen, 2007*).
- Similarity measures between IFS (*Chen & Chang, 2015*).

However, IFS has limitations when the sum of satisfactory and non-satisfactory grades equals 1, rendering it less effective in specific decision-making scenarios. To overcome this challenge, Yager introduced the concept of PFS, characterized by the sum of the squares of satisfactory and non-satisfactory grades falling within the range of [0,1]. PFS proved to be a more suitable choice than IFS for dealing with uncertain and vague data in decision-making, enhancing the reliability of the decision-making process.

Yager further expanded upon PFS by introducing the q-rung orthopair fuzzy set (q-ROFS). q-ROFS represents an advanced form of both IFS and PFS, allowing for the sum of the qth power of satisfactory and non-satisfactory grades, constrained within the range of 1. In q-ROFS, q serves as a parameter, affording a comprehensive framework for describing uncertain and vague data regarding satisfactory and non-satisfactory grades (*Yager & Alajlan, 2017*). Future advancements can be made in the following directions as well to enhance the literature broadness. The proposed work can be extended to develop a sliding mode control strategy for discrete-time interval type-2 fuzzy Markov jump systems with preview target signals (*Sun, Ren & Zhao, 2022*), an adaptive inventory control model using fuzzy neural networks in uncertain environments (*Ge & Zhang, 2019*), and an observer-based sliding mode control method for fuzzy stochastic switching systems facing deception attacks (*Zhang et al., 2022*). It can be extended to explore fuzzy fixed-point results applied to fuzzy differential equations (*Sarwar & Li, 2019*), fuzzy sampled-data

stabilization for chaotic nonlinear systems (*Xia et al., 2020*) and focus on sliding mode control for semi-Markov jump T-S fuzzy systems with time delays (*Gao et al., 2020*).

The q-rung ortopair fuzzy values contains more information in comparison of fuzzy values. The AHP approach was introduced in the environment of a q-rung orthopair double hierarchy linguistic term set, and its applications were explored in security systems (*Duan et al., 2022*). Secondly, the complex information can efficiently be handle using q-rung orthopair fuzzy set as it reduces information loss. In a complex MCDM problem, the weights for criteria plays very important role and the fuzzy AHP can handle such situation due to flexibility of assigning weights. Thus q-rung ortopair fuzzy AHP becomes an effective platform for handling complex decision making probems as it combines the characteristics of fuzzy AHP and q-rung orthopair fuzzy set. This article delves into fuzzy AHP within the q-rung orthopair fuzzy sets framework, introducing triangular q-rung orthopair fuzzy numbers (Trq-ROFNs). Our methodology includes defining Tr-q-ROFNs scores and accuracy functions, facilitating the selection of the best coffee brand through an innovative q-rung orthopair fuzzy AHP approach. Additionally, we compare our results with existing methodologies to validate the accuracy of our proposed technique.

## PRELIMINARIES

The decision making is one of fundamental and critical problem in real life. In a complex decision making problem there always been a need to handle ambiguity and information loss. Here, we present some definitions and the results of fuzzy sets and q-ROFSs.

Definition 1 (*Zadeh, 1965*) A function $S$ from the universal set $R$ to close interval $[0,1], S \; : \; R \to [0,1]$ is called fuzzy set (FS). The $S(x)$ for $x \in R$ is called a satisfactory grade of $x$.

Definition 2 (*Atanassov, 2012*) A function $S$ from the universal set $R$ to close interval $[0,1], S \; : \; R \to [0,1]$ of the form $S(r) = \langle \breve{E}_S(r), \dot{G}_S(r) \rangle$, for all $r \in R$, were $\breve{E}_S(r)$ and $\dot{G}_S(r)$ are satisfactory grade and non-satisfactory grade of $r$, with the condition, $0 \leq \breve{E}(r) \leq 1, 0 \leq \dot{G}_S(r) \leq 1$ for all $r \in R$ and $0 \leq \breve{E}_s(r) + \dot{G}_S(r) \leq 1$ for all $r \in R$ and for all $r \in R$ is called intuitionistic fuzzy set (IFS). The hesitancy grade or degree measure is followed as $\Pi_S(r) = 1 - \breve{E}_s(r) - \dot{G}_S(r)$ for all $r \in R$.

Definition 3 (*Yager, 2014*) Let $R$ be nonempty set. A Pythagorean fuzzy set (PFS) is the structure of the form $S(r) = \langle \breve{E}_S(r), \dot{G}_S(r) \rangle$ for all $r \in R$, where $\breve{E}_S(r)$ and $\dot{G}_S(r)$ are satisfactory and non-satisfactory grades of $r$ in $R$ PFS $S$ with the condition $0 \leq \breve{E}_S(r) \leq 1, 0 \leq \dot{G}_S(r) \leq 1$ for all $r \in R$ and $0 \leq \left(\breve{E}_S(r)\right)^2 + \left(\dot{G}_S(r)\right)^2 \leq 1.$ *for all* $r \in R$. For each PFS, $S$ in $R$ the hesitancy grade or degree measure is given by

$\Pi_S(r) = \sqrt{1 - \left(\breve{E}_S(r)\right)^2 + \left(\dot{G}_S(r)\right)^2}$, for all $r \in R$.

Definition 4 (*Yager & Alajlan, 2017*) Let $R$ be a nonempty set. A q-rung orthopair fuzzy set (q-ROFS) is the structure of the form, $S = \{r, \breve{E}_S(r), \dot{G}_S(r) \,|\, r \in R\}$, where $\breve{E}_S : r \to [0,1]$ and $\dot{G}_S \; : \; r \to [0,1]$ are called satisfactory and non- satisfactory grades, respectively, with the condition, $0 \leq \breve{E}_S(r)^q + \dot{G}_S(r)^q \leq 1, q \geq 1$, for all $r \in R$. The degree of indeterminacy of the element $r \in R$ is given as $\Pi_S(r) = \langle \breve{E}_S(r)^q + \dot{G}_S(r)^q - \breve{E}_S(r)^q \dot{G}_S(r)^q \rangle^{\frac{1}{q}}$. For convenience, we denote $(\breve{E}_s(r), \dot{G}_S(r))$ by $S = (\breve{E}, \dot{G})$.

Definition 5 (*Yager & Alajlan, 2017*) Let $\alpha = (\breve{E}, \dot{G}), \alpha_1 = (\breve{E}_1, \dot{G}_1), \alpha_2 = (\breve{E}_2, \dot{G}_2)$, be any three q-ROFNs. Then, the operations are defined as

1. $\alpha_1 \vee \alpha_2 = \langle max\{\breve{E}_1, \breve{E}_2\}, min\{\dot{G}_1, \dot{G}_2\}\rangle$,

2. $\alpha_1 \wedge \alpha_2 = \langle min\{\breve{E}_1, \breve{E}_2\}, max\{\dot{G}_1, \dot{G}_2\}\rangle$,

3. $\alpha_1 \oplus \alpha_2 = \langle\langle \breve{E}_1^q + \breve{E}_2^q - \breve{E}_1^q \breve{E}_2^q\rangle^{\frac{1}{q}}, \dot{G}_1, \dot{G}_2\rangle$,

4. $\alpha_1 \otimes \alpha_2 = \langle \breve{E}_1 \breve{E}_2, \langle \dot{G}_1^q + \dot{G}_2^q - \dot{G}_1^q \dot{G}_2^q\rangle^{\frac{1}{q}}\rangle$,

5. $\lambda\alpha_1 = \left(\left(1 - \langle 1 - \breve{E}_1^q\rangle^{\lambda}\right)^{\frac{1}{q}}, \dot{G}_1^{\lambda}\right)$,

6. $\alpha_1^{\lambda} = \left(\breve{E}_1^{\lambda}, \left(1 - \langle 1 - \dot{G}_1^q\rangle^{\lambda}\right)^{\frac{1}{q}}\right)$.

## THE Q-RUNG ORTHOPAIR FUZZY ANALYTIC HIERARCHY PROCESS

In this section, we propose a novel approach to the analytic hierarchy process under the environment of a q-rung ortho pair fuzzy set. First, we define the concept of Tr-q-ROFNs with some basic operations.

Definition 6 (*Fahmi & Aslam, 2021*) $L = \left[(l, m, n), \breve{E}, \dot{G}\right]$ Let L = be a Tr-q-ROFN. We define.

$$R_{\breve{E}(x)} = \begin{cases} \dfrac{x - l}{m - l}(\breve{E}), & l \leq x < m \\ \dfrac{n - x}{n - m}(\breve{E}), & m \leq x \leq n \\ 0, & \text{otherwise} \end{cases}$$

and

$$R_{\dot{G}(x)} = \begin{cases} \dfrac{x - l}{m - l}(\dot{G}), & l \leq x < m \\ \dfrac{n - x}{n - m}(\dot{G}), & m \leq x \leq n \\ 0, & \text{otherwise} \end{cases}$$

where $R_{\breve{E}(x)}$ is satisfactory and $R_{\dot{G}(x)}$ is the non-satisfactory grade of $\left[(l, m, n), \breve{E}, \dot{G}\right]$.

Definition 7 (*Fahmi & Aslam, 2021*) Let $\tilde{a}_1 = \langle(a_1, a_2, a_3), E_{a1}, G_{a1}\rangle$ and $\tilde{a}_2 = \langle(b_1, b_2, b_3), E_{a2}, G_{a2}\rangle$ be two Tr-q-ROFNs. Then

1-Addition of two triangular q-ROFNs;

$\tilde{a}_1 + \tilde{a}_2 = \langle(a_1 + b_1, a_2 + b_2, a_3 + b_3), \langle E_{a_1}^q + E_{a_1}^q - E_{a_1}^q E_{a_1}^q\rangle^{\frac{1}{q}}, G_{\tilde{a}1} G_{\tilde{a}_2}\rangle$,

2-Multiplication of two triangular q-ROFNs;

$\tilde{a}_1 \times \tilde{a}_2 = \left((a_1 b_1, a_2 b_2, a_3 b_3), E_{a1} E_{a2}, \langle G_{a1}^q + G_{a2}^q - G_{\tilde{a}_1}^q G_{\tilde{a}_2}^q\rangle^{\frac{1}{q}}\right)$,

3-Multiplication of a triangular q-ROFN by scalar.

$\eta\tilde{a}_1 = \langle(\eta a_1, \eta a_2, \eta a_3), \langle 1 - \langle 1 - E_{a1}^q\rangle^{\eta}\rangle^{\frac{1}{q}}\rangle$,

4-Inverse of triangular q-ROFN;

$$\tilde{a}_1^{-1} = \left\langle \left(\frac{1}{a_3}, \frac{1}{a_2}, \frac{1}{a_1}\right), E_{a1}, G_{a1} \right\rangle.$$

**Definition 8** (*Fahmi & Aslam, 2021*) The score function of triangular q-rung orthopair fuzzy numbers;

$$S(\breve{a}) = \frac{(\breve{a}_1 + \overline{a}_2 + \breve{a}_3)(1 + E_1^q - G_1^q)}{6}$$

and the accuracy function of triangular q-ROFNs:

$$H(\breve{a}) = \frac{(\breve{a}_1 + \breve{a}_2 + \breve{a}_3)(1 + E_1^q + G_1^q)}{6}$$

### q-rung orthopair fuzzy analytic hierarchy process

The expert's opinion plays a pivotal role in an MCDM problem as the information relies heavily on data given by experts. The q-rung orthopair fuzzy set can efficiently handle the uncertainty and minimize information loss in a complex situation. The fuzzy AHP is used in an MCDM problem due to flexibility of assigning weights. Thus q-rung ortopair fuzzy AHP will provide an efficient and reliable platform for decision making as it combines the characteristics of fuzzy AHP and q-rung orthopair fuzzy set.

Next we discuss the different steps for the construction of q-rung orthopair FAHP. The method is effective for handling both qualitative and quantitave data. For qualitative data, the linguistic terms are converted into q-rung orthopair fuzzy values using evaluation scale presented in Table 1. For quantitative data, the crisp data is converted to q-rung orthopair fuzzy values and then AHP method is applied.

**Step 1.** First, we construct the hierarchical structure of the problem. This hierarchical structure consists of four stages, (i) Choosing the desirable goal, (ii) Criteria (iii) Sub criteria (iv) ranking of alternatives. Hierarchal structure of q-rung orthopair fuzzy AHP is shown in Fig. 1.

**Step 2.** In this step, we construct the pairwise comparison matrix of criteria, sub-criteria, and alternatives, and their linguistic triangular scale as shown in Table 1.

The q-rung ortho pair triangular fuzzy scale is based on the expert's opinion. The q-rung ortho pair-wise comparison matrix of criteria, sub-criteria, and alternative are as follows,

$$A = \begin{bmatrix} \tilde{a_{11}} & \tilde{a_{12}} & . & . & \tilde{a_{1n}} \\ \tilde{a_{21}} & \tilde{a_{22}} & . & . & . \\ . & . & . & . & . \\ . & . & . & . & . \\ \tilde{a_{n1}} & \tilde{a_{n2}} & . & . & \tilde{a_{mn}} \end{bmatrix}$$

where $\tilde{a_{ji}} = \tilde{a_{ij}}^{-1}$ is a q-rung ortho pair triangular fuzzy number that is used to measure the indeterminacy in decision.

**Table 1 Linguistic terms evaluation scale.**

| Saaty scale | Explanation | q-rung orthopair triangular scale |
|---|---|---|
| 1 | Identical | $\tilde{1} = \langle(1,1,1), 0.50, 0.50\rangle$ |
| 3 | Important | $\tilde{3} = \langle(2,3,4), 0.30, 0.70\rangle$ |
| 5 | Very important | $\tilde{5} = \langle(4,5,6), 0.80, 0.20\rangle$ |
| 7 | Very very important | $\tilde{7} = \langle(6,7,8), 0.70, 0.30\rangle$ |
| 9 | Perfect | $\tilde{9} = \langle(9,9,9), 1.0, 0.1\rangle$ |
| 2 | Middle values between two closest scales | $\tilde{2} = \langle(1,2,3), 0.40, 0.60\rangle$ |
| 4 | | $\tilde{4} = \langle(3,4,5), 0.50, 0.60\rangle$ |
| 6 | | $\tilde{6} = \langle(5,6,7), 0.70, 0.30\rangle$ |
| 8 | | $\tilde{8} = \langle(7,8,9), 0.85, 0.15\rangle$ |

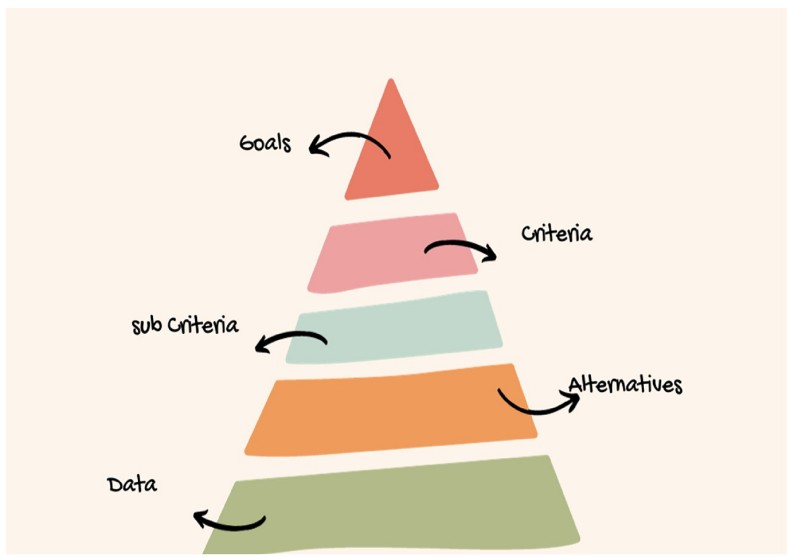

**Figure 1 Hierarchal structure of q-rung orthopair fuzzy AHP.**

**Step 3.** To check the consistency of the expert's judgment. If the pair-wise comparison matrix is consistent, then we have $a_{ik} = a_{ij}a_{jk}$ for $i, j, k$. In triangular q-rung ortho pair triangular fuzzy number of the comparison matrix, important are its upper, middle, and lower values.

**Step 4.** In this step, we use q-rung ortho pair-wise comparison matrix to analyze the weight of alternatives, criteria, and sub-criteria. By using operations of q-ROFSs we normalized the data of rows.

$$w_j = \frac{\sum_{j=1}^n a_{ij}}{\sum_{i=1}^m \sum_{j=1}^n a_{ij}}$$

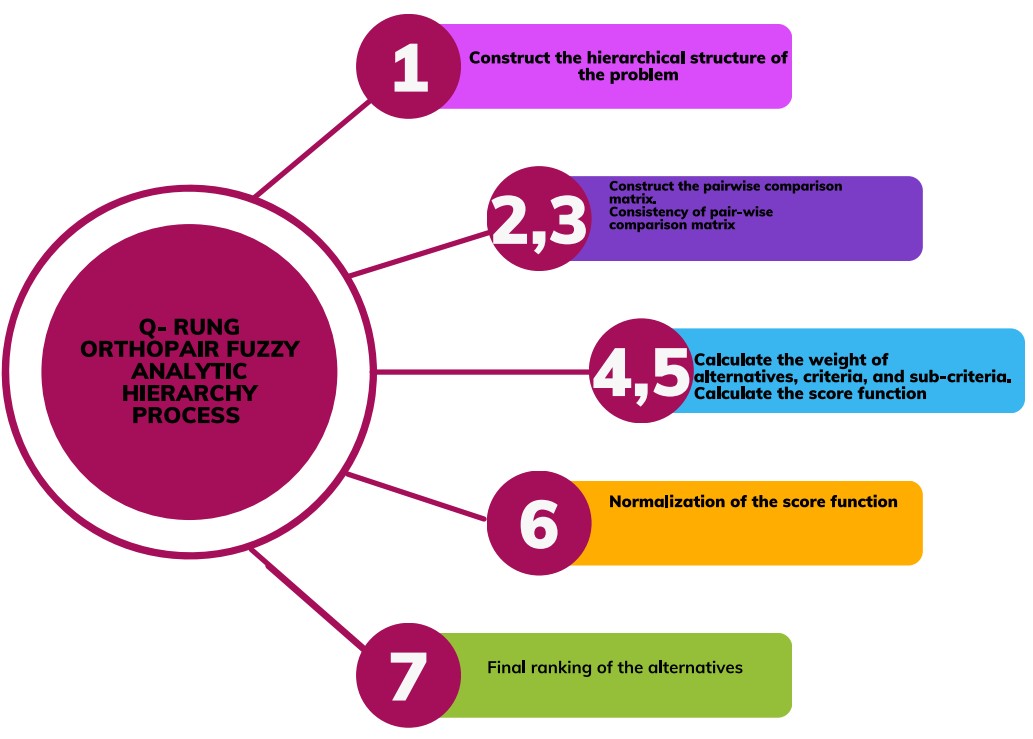

**Figure 2** Flowchart. 

where

$$\sum_{i=1}^{m}\sum_{j=1}^{n} a_{ij} = \left(\sum_{i=1}^{m}\sum_{j=1}^{n} a_{ij}, \sum_{i=1}^{m}\sum_{j=1}^{n} b_{ij}, \sum_{i=1}^{m}\sum_{j=1}^{n} c_{ij}\right)$$

and $(i = 1, 2, 3 \ldots m), (j = 1, 2, 3 \ldots n)$

**Step 5.** Let $a_{\tilde{ij}} = \langle (\breve{a}_1, \breve{a}_2, \breve{a}_3), (E_a, G_a) \rangle$ be the triangular q-ROFNs, then

$$\hat{S}\left(a_{\widetilde{ij}}\right) = \frac{(a_1 + a_2 + a_3)(1 + (E_a - G_a))}{3} \text{ and } \breve{A}\left(\tilde{a}_{ij}\right) = \frac{(a_1 + a_2 + a_3)(1 + E_a + G_a)}{3}$$

where $\hat{S}(\alpha)$ and $\tilde{A}(\alpha)$ is score and accuracy function.

**Step 6.** Normalize the score function (weight vector) by adding the rows and dividing the total of the row by each entry.

**Step 7.** For the final ranking of alternatives, we use the above relation, and $j = 1, 2, 3 .. n$. Then we get the results of the analytic hierarchy process under the environment of q-ROFSs as given below as flow chart is given in Fig. 2.

If we want to calculate the score function of $\hat{S}\left(a_{\widetilde{ij}}\right)$ the formula is given below as;

$$\hat{S}\left(a_{\widetilde{ij}}\right) = \frac{1}{\hat{S}\left(a_{\widetilde{ij}}\right)}$$

## NUMERICAL APPLICATIONS

Coffee has become an essential product in international trade. The world's demand for coffee has increased in the last decade. With the increase in the need for coffee, the

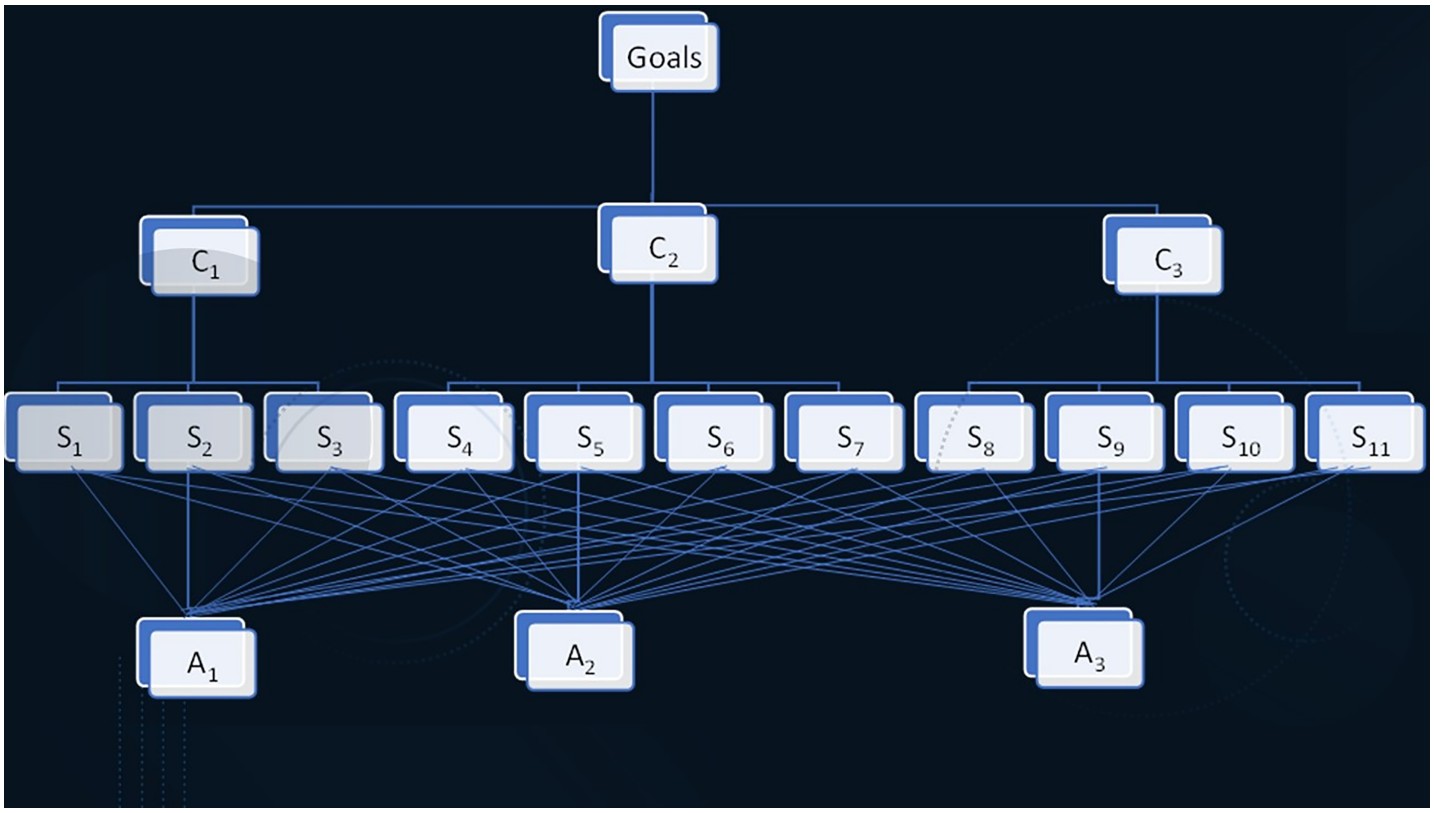

**Figure 3  Hierarchy structure.**               

consumption of coffee has also increased very much. When different brands of coffee are available in the markets, the decision maker has alternatives to select the best coffee brand. A complex set of factors, like the product quality, convenience, brewing method, and pricing determines the final decision. The sub-factors are given as follows:

For the product quality, sub-factors are certifications, coffee bean type, freshness, roast level and speciality coffee. The sub-factors for the convenience are packaging, preparation time, availability, online ordering and portability. Most commonly used brewing methods are aeropress, chemex, and cold brew. The pricing includes coffee bean price, discounts, value for money, premium pricing and price comparison.

Fuzzy AHP is a structural technique that gives the best ranking result under the environment of a q-rung orthopair fuzzy set. The novel approach of q-rung fuzzy AHP is the best way of decision analysis. For this reason, we use q rung fuzzy AHP to select the best coffee brand. Here, we use q rung fuzzy triangular numbers to find the final ranking.

**Example 1**: For a man who wants to buy coffee in South Africa, he can consider three different coffee brands: Assagay Coffee, Douwe Egberts, and Jacobs Kronung. Multiple factors affect his brand selection decision, including social, cultural, and personal factors. This study considers these factors as criteria, and sub-criteria include $S_1$ family choice, $S_2$ reference, $S_3$ status, $S_4$ age factor, $S_5$ income of DMS, $S_6$ lifestyle, $S_7$ personality, $S_8$ occupation, $S_9$ culture, $S_{10}$ subculture, and $S_{11}$ social class. Using q-rung fuzzy AHP, the

**Table 2 The q-rung orthopair comparison matrix for criteria.**

| Criteria | $C_1$ | $C_2$ | $C_3$ | q-rung ortho-pair | weights | Normalized weights |
|---|---|---|---|---|---|---|
| $C_1$ | $\tilde{1}$ | $\tilde{3}$ | $\tilde{7}^{-1}$ | $\langle(0.12, 0.16, 0.22), 0.04, 0.76\rangle$ | 0.093 | 0.12 |
| $C_2$ | $\tilde{3}^{-1}$ | $\tilde{1}$ | $\tilde{5}^{-1}$ | $\langle(0.06, 0.09, 0.20), 0.03, 0.79\rangle$ | 0.059 | 0.08 |
| $C_3$ | $\tilde{7}$ | $\tilde{5}$ | $\tilde{1}$ | $\langle(0.57, 0.73, 0.94), 0.09, 0.56\rangle$ | 0.615 | 0.80 |

**Table 3 The q-rung orthopair comparison matrix with respect to C1.**

| Criteria | $S_1$ | $S_2$ | $S_3$ | Priority vector |
|---|---|---|---|---|
| $S_1$ | $\tilde{1}$ | $\tilde{3}$ | $\tilde{7}^{-1}$ | 0.75 |
| $S_2$ | $\tilde{3}^{-1}$ | $\tilde{1}$ | $\tilde{5}^{-1}$ | 0.13 |
| $S_3$ | $\tilde{7}$ | $\tilde{5}$ | $\tilde{1}$ | 0.11 |

**Table 4 The q-rung orthopair comparison matrix with respect to C2.**

| Criteria | $S_4$ | $S_5$ | $S_6$ | $S_7$ | Priority vector |
|---|---|---|---|---|---|
| $S_4$ | $\tilde{1}$ | $\tilde{5}$ | $\tilde{7}^{-1}$ | $\tilde{7}$ | 0.23 |
| $S_5$ | $\tilde{5}^{-1}$ | $\tilde{1}$ | $\tilde{7}^{-1}$ | $\tilde{9}$ | 0.10 |
| $S_6$ | $\tilde{7}$ | $\tilde{7}$ | $\tilde{1}$ | $\tilde{7}$ | 0.63 |
| $S_7$ | $\tilde{7}^{-1}$ | $\tilde{9}^{-1}$ | $\tilde{7}^{-1}$ | $\tilde{1}$ | 0.02 |

**Table 5 The q-rung orthopair comparison matrix w. r. to C3.**

| Criteria | $S_8$ | $S_9$ | $S_{10}$ | $S_{11}$ | Priority vector |
|---|---|---|---|---|---|
| $S_8$ | $\tilde{1}$ | $\tilde{9}^{-1}$ | $\tilde{9}$ | $\tilde{7}^{-1}$ | 0.08 |
| $S_9$ | $\tilde{9}$ | $\tilde{1}$ | $\tilde{9}$ | $\tilde{7}$ | 0.66 |
| $S_{10}$ | $\tilde{9}^{-1}$ | $\tilde{9}^{-1}$ | $\tilde{1}$ | $\tilde{7}^{-1}$ | 0.02 |
| $S_{11}$ | $\tilde{7}^{-1}$ | $\tilde{7}^{-1}$ | $\tilde{7}$ | $\tilde{1}$ | 0.22 |

decision makers can choose the best coffee brand based on the above-mentioned factors. Let us consider three alternatives having a hierarchy decomposition structure, as shown in Fig. 3.

This structure has four stages: goal, criteria, sub-criteria, and alternatives. Construct the decision pair-wise q-rung ortho-pair comparison matrix in linguistic terms using above mention criteria and alternatives. The q-rung ortho pair-wise comparison matrix for criteria is given as follows in Table 2.

The q-rung ortho pair-wise comparison matrix with respect to $C_1$, in Table 3 and its priority vector.

**Table 6 The q-rung orthopair comparison matrix with respect to S1, S2, S3.**

| $S_1$ | $A_1$ | $A_2$ | $A_3$ | Priority vector | $S_2$ | $A_1$ | $A_2$ | $A_3$ | Priority vector | $S_3$ | $A_1$ | $A_2$ | $A_3$ | Priority vector |
|---|---|---|---|---|---|---|---|---|---|---|---|---|---|---|
| $A_1$ | $\tilde{1}$ | $\tilde{7}$ | $\tilde{5}$ | 0.69 | $A_1$ | $\tilde{1}$ | $\tilde{3}$ | $\tilde{9}^{-1}$ | 0.08 | $A_1$ | $\tilde{1}$ | $\tilde{3}$ | $\tilde{7}^{-1}$ | 0.11 |
| $A_2$ | $\tilde{7}^{-1}$ | $\tilde{1}$ | $\tilde{5}^{-1}$ | 0.06 | $A_2$ | $\tilde{3}^{-1}$ | $\tilde{1}$ | $\tilde{5}^{-1}$ | 0.04 | $A_2$ | $\tilde{3}^{-1}$ | $\tilde{1}$ | $\tilde{5}^{-1}$ | 0.02 |
| $A_3$ | $\tilde{5}^{-1}$ | $\tilde{5}$ | $\tilde{1}$ | 0.24 | $A_3$ | $\tilde{9}$ | $\tilde{5}$ | $\tilde{1}$ | 0.87 | $A_3$ | $\tilde{7}$ | $\tilde{5}$ | $\tilde{1}$ | 0.82 |

**Table 7 The q-rung orthopair comparison matrix with respect to S4, S5, S6, S7.**

| $S_4$ | $A_1$ | $A_2$ | $A_3$ | PV | $S_5$ | $A_1$ | $A_2$ | $A_3$ | PV | $S_6$ | $A_1$ | $A_2$ | $A_3$ | PV | $S_7$ | $A_1$ | $A_2$ | $A_3$ | PV |
|---|---|---|---|---|---|---|---|---|---|---|---|---|---|---|---|---|---|---|---|
| $A_1$ | $\tilde{1}$ | $\tilde{7}^{-1}$ | $\tilde{3}$ | 0.19 | $A_1$ | $\tilde{1}$ | $\tilde{7}$ | $\tilde{3}$ | 0.67 | $A_1$ | $\tilde{1}$ | $\tilde{9}^{-1}$ | $\tilde{7}^{-1}$ | 0.05 | $A_1$ | $\tilde{1}$ | $\tilde{3}$ | $\tilde{1}$ | 0.27 |
| $A_2$ | $\tilde{7}$ | $\tilde{1}$ | $\tilde{3}$ | 0.74 | $A_2$ | $\tilde{7}^{-1}$ | $\tilde{1}$ | $\tilde{1}$ | 0.18 | $A_2$ | $\tilde{9}$ | $\tilde{1}$ | $\tilde{7}^{-1}$ | 0.18 | $A_2$ | $\tilde{3}^{-1}$ | $\tilde{1}$ | $\tilde{7}^{-1}$ | 0.08 |
| $A_3$ | $\tilde{3}^{-1}$ | $\tilde{3}^{-1}$ | $\tilde{1}$ | 0.05 | $A_3$ | $\tilde{3}^{-1}$ | $\widetilde{1}$ | $\tilde{1}$ | 0.13 | $A_3$ | $\tilde{7}$ | $\tilde{7}$ | $\tilde{1}$ | 0.64 | $A_3$ | $\tilde{1}$ | $\tilde{7}$ | $\tilde{1}$ | 0.64 |

**Table 8 The q-rung orthopair comparison matrix with respect to S8, S9, S10, S11.**

| $S_8$ | $A_1$ | $A_2$ | $A_3$ | PV | $S_9$ | $A_1$ | $A_2$ | $A_3$ | PV | $S_{10}$ | $A_1$ | $A_2$ | $A_3$ | PV | $S_{11}$ | $A_1$ | $A_2$ | $A_3$ | PV |
|---|---|---|---|---|---|---|---|---|---|---|---|---|---|---|---|---|---|---|---|
| $A_1$ | $\tilde{1}$ | $\tilde{9}$ | $\tilde{7}$ | 0.82 | $A_1$ | $\tilde{1}$ | $\tilde{5}$ | $\tilde{7}^{-1}$ | 0.29 | $A_1$ | $\tilde{1}$ | $\tilde{9}$ | $\tilde{3}$ | 0.76 | $A_1$ | $\tilde{1}$ | $\tilde{1}$ | $\tilde{9}^{-1}$ | 0.08 |
| $A_2$ | $\widetilde{9}^{-1}$ | $\tilde{1}$ | $\tilde{1}$ | 0.08 | $A_2$ | $\tilde{5}^{-1}$ | $\tilde{1}$ | $\tilde{9}$ | 0.47 | $A_2$ | $\tilde{9}^{-1}$ | $\tilde{1}$ | $\tilde{3}$ | 0.17 | $A_2$ | $\tilde{1}$ | $\tilde{1}$ | $\tilde{7}^{-1}$ | 0.08 |
| $A_3$ | $\tilde{7}^{-1}$ | $\tilde{1}$ | $\tilde{1}$ | 0.08 | $A_3$ | $\tilde{7}$ | $\tilde{9}^{-1}$ | $\tilde{1}$ | 0.23 | $A_3$ | $\tilde{3}^{-1}$ | $\tilde{3}^{-1}$ | $\tilde{1}$ | 0.05 | $A_3$ | $\tilde{9}$ | $\tilde{7}$ | $\tilde{1}$ | 0.82 |

**Table 9 The final weight of sub criteria's, final weight vector with respect to C1.**

| | $S_1$ | $S_2$ | $S_3$ | Local weight |
|---|---|---|---|---|
| Weight | 0.75 | 0.13 | 0.11 | |
| $A_1$ | 0.69 | 0.08 | 0.11 | 0.54 |
| $A_2$ | 0.06 | 0.04 | 0.06 | 0.02 |
| $A_3$ | 0.24 | 0.87 | 0.82 | 0.39 |

The q-rung ortho pair-wise comparison matrix with respect to $C_2$, in Table 4 and its priority vector.

In Table 5, the q-rung ortho pair-wise comparison matrix w. r. to $C_3$, and its final vectors are given.

In Table 6, the q-rung ortho pair-wise comparison matrix with respect to $S_1, S_2, S_3$ and its final vectors are given.

In Table 7, the q-rung ortho pair-wise comparison matrix with respect to $S_4, S_5, S_6$ and $S_7$ and its final vectors are given.

In Table 8, the q-rung ortho pair-wise comparison matrix with respect to $S_8, S_9, S_{10}$ and $S_{11}$ and its final vectors are given.

**Table 10 Final weight vector with respect to C2.**

|  | $S_4$ | $S_5$ | $S_6$ | $S_7$ | Local weight |
|---|---|---|---|---|---|
| Weight | 0.23 | 0.10 | 0.63 | 0.02 | |
| $A_1$ | 0.19 | 0.67 | 0.05 | 0.27 | 0.15 |
| $A_2$ | 0.74 | 0.18 | 0.18 | 0.08 | 0.30 |
| $A_3$ | 0.05 | 0.13 | 0.64 | 0.64 | 0.44 |

**Table 11 Final weight vector with respect to C3.**

|  | $S_8$ | $S_9$ | $S_{10}$ | $S_{11}$ | Local weight |
|---|---|---|---|---|---|
| Weight | 0.08 | 0.66 | 0.02 | 0.22 | |
| $A_1$ | 0.82 | 0.29 | 0.76 | 0.08 | 0.29 |
| $A_2$ | 0.08 | 0.47 | 0.17 | 0.08 | 0.34 |
| $A_3$ | 0.08 | 0.23 | 0.05 | 0.82 | 0.35 |

**Table 12 Final ranking of alternatives with respect to a goal.**

|  | $C_1$ | $C_2$ | $C_3$ | Global weight | Ranking |
|---|---|---|---|---|---|
| Weight | 0.12 | 0.08 | 0.80 | | |
| $A_1$ | 0.54 | 0.15 | 0.29 | 0.31 | 2nd |
| $A_2$ | 0.02 | 0.30 | 0.34 | 0.30 | 3rd |
| $A_3$ | 0.39 | 0.44 | 0.35 | 0.36 | 1st |

Now the final weight of sub criteria's, final weight vector with respect to $C_1$ are given in Table 9.

Final weight vector with respect to $C_2$ are given in Table 10.

Final weight vector with respect to $C_3$ are given in Table 11.

Final ranking of alternatives with respect to a goal are given in Table 12.

Hence the final ranking result of the best alternatives are, $A_3 \succ A_1 \succ A_2$. The best coffee brand for a decision maker is $A_3$ based on the social, cultural, and personal factors. The best result, as shown alternatives along y-axis and resulting values along x-axis is shown in Fig. 4 as below,

**Example 2**: For the selection of the best coffee brand, we consider three different coffee brands $A_1$, $A_2$, and $A_3$, as an alternative based on the following criteria $C_1$; Availability, $C_2$; effectiveness, $C_3$; price, $C_4$; quality, $C_5$; quantity. First, let us consider three alternatives having a hierarchical decomposition structure, as shown in figure (a). This structure has four stages Goal, Criteria, Sub-criteria, and Alternatives. Then we construct the pair-wise q-rung ortho-pair comparison matrix in linguistic terms using the criteria mentioned earlier and alternatives as given in Tables 1 and 2 employing the expert opinions.

To construct hierarchy structure of the problem as in Fig. 3.

The q-rung ortho pair wise comparison matrix with respect to criteria is given by

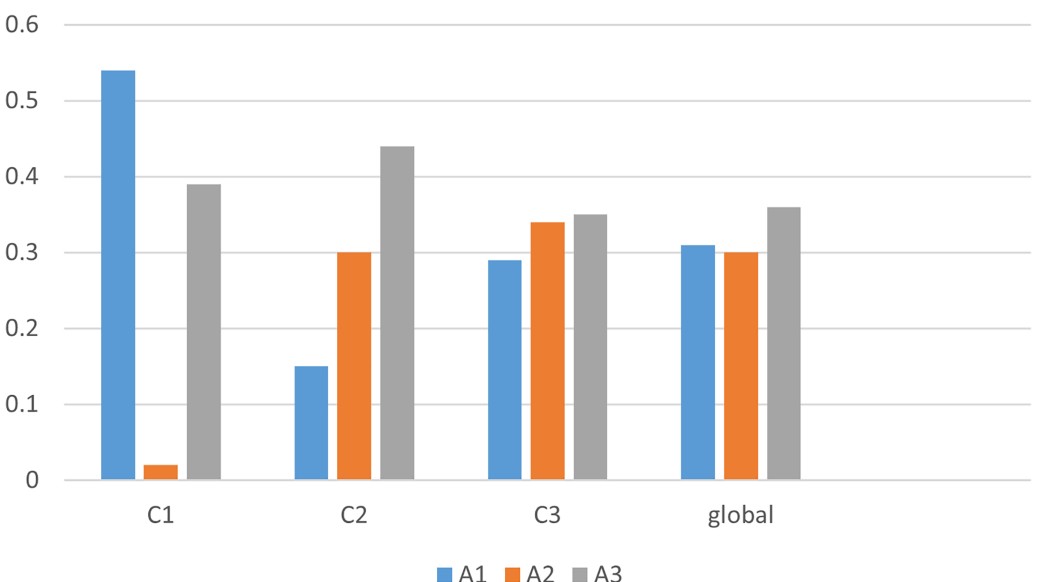

**Figure 4 Graphical view of the best alternative based on score function.**

$$
\begin{bmatrix}
1^{\sim} & 2^{\sim} & 4^{\sim} & 9^{\sim} & 5^{\sim} \\
2^{\sim-1} & 1^{\sim} & 6^{\sim} & 7^{\sim} & 2^{\sim} \\
4^{\sim-1} & 7^{\sim-1} & 1^{\sim} & 5^{\sim} & 3^{\sim} \\
9^{\sim-1} & 6^{\sim-1} & 5^{\sim-1} & 1^{\sim} & 2^{\sim} \\
5^{\sim-1} & 2^{\sim-1} & 3^{\sim-1} & 2^{\sim-1} & 1^{\sim}
\end{bmatrix}
$$

The deterministic q-rung ortho pair-wise comparison matrix w.r.to criteria's, using score function, the above q-rung ortho pair-wise comparison matrix transformed into deterministic pair-wise comparison matrix as

$$
\begin{bmatrix}
Criteria & C_1 & C_2 & C_3 & C_4 & C_5 \\
C_1 & 1 & 1.86 & 4.27 & 17.99 & 5.82 \\
C_2 & 0.56 & 1 & 11.13 & 7.96 & 1.86 \\
C_3 & 0.27 & 0.23 & 1 & 5.82 & 1.50 \\
C_4 & 0.22 & 0.22 & 0.23 & 1 & 1.86 \\
C_5 & 0.23 & 0.56 & 0.90 & 0.56 & 1
\end{bmatrix}
$$

Then calculating the rank of criteria's expressing the eigen vector $X$, then proceeding matrix, as demonstrated earlier in the extensive steps of the estimated form. Then normalizing pair-wise comparison matrix obtain. Normalized pair-wise comparison matrix of criteria is

$$
\begin{bmatrix}
0.43 & 0.48 & 0.24 & 0.05 & 0.44 \\
0.24 & 0.25 & 0.63 & 0.23 & 0.14 \\
0.11 & 0.05 & 0.05 & 0.17 & 0.19 \\
0.09 & 0.05 & 0.01 & 0.03 & 0.14 \\
0.10 & 0.14 & 0.5 & 0.01 & 0.07
\end{bmatrix}
$$

Now take the total of row averages.

$$X = \begin{bmatrix} 0.42 \\ 0.29 \\ 0.12 \\ 0.07 \\ 0.08 \end{bmatrix}$$

**Step 3:** It is important to know that data is consistent by using consistency test, and get consistency test as follows:

$$\begin{bmatrix} 1 & 1.86 & 4.27 & 17.99 & 5.82 \\ 0.56 & 1 & 11.13 & 7.96 & 1.86 \\ 0.27 & 0.23 & 1 & 5.82 & 2.50 \\ 0.22 & 0.22 & 0.23 & 1 & 1.86 \\ 0.23 & 0.56 & 0.90 & 0.56 & 1 \end{bmatrix} . \begin{bmatrix} 0.42 \\ 0.29 \\ 0.12 \\ 0.07 \\ 0.08 \end{bmatrix} = \begin{bmatrix} 3.10 \\ 2.50 \\ 0.87 \\ 0.39 \\ 0.48 \end{bmatrix}$$

From the above equations

$$\lambda \text{average} \left\{ \frac{3.10}{0.42}, \frac{2.50}{0.29}, \frac{0.87}{0.12}, \frac{0.39}{0.07}, \frac{0.48}{0.08} \right\}_{max}$$

and $Cι = \dfrac{\phi_{max}}{n-1} \dfrac{5.28}{5-1} = \dfrac{0.28}{4} = 0.07$. For consistency ratio we take the values table from

*Saaty (1990a, 1998, 1990b)* as

$$\begin{bmatrix} 1 & 2 & 3 & 4 & 5 & 6 & 7 & 8 & 9 & 10 \\ 0 & 0 & 0.58 & 0.90 & 1.12 & 1.24 & 1.32 & 1.4 & 1.45 & 1.49 \end{bmatrix}$$

$$CR = \frac{CI}{RI}$$
$$= \frac{0.070}{1.12} = 0.0625. = 6.2\%.$$

The evaluation is consistent as 6.2% < 10%.

The q-rung orthopair comparison matrix, normalized decision matrix and vector $X$ w.r. to $C_1$ is given in Table 13.

he q-rung orthopair comparison matrix, normalized decision matrix and vector $X$ w.r.to $C_2$ is given in Table 14.

The q-rung orthopair comparison matrix, normalized decision matrix and vector $X$ w.r. to $C_3$ is given in Table 15.

The q-rung orthopair comparison matrix, normalized decision matrix and vector $X$ w.r. to $C_4$ is given in Table 16.

The q-rung orthopair comparison matrix, normalized decision matrix and vector $X$ w.r. to $C_5$ is given in Table 17.

The final ranking of alternatives is given below in Table 18 with respect to goal.

**Table 13 The q-rung orthopair comparison matrix, normalized decision matrix and vector X w.r.t to C1.**

| q-rung orthopair decision matrix | $A_1$ | $A_2$ | $A_3$ | Decision matrix | $A_1$ | $A_2$ | $A_3$ | Normalized decision matrix | $A_1$ | $A_2$ | $A_3$ | Vector X |
|---|---|---|---|---|---|---|---|---|---|---|---|---|
| | | | | | | | | | | | | |
| $A_1$ | $\tilde{1}$ | $\tilde{5}^{-1}$ | $\tilde{2}^{-1}$ | $A_1$ | 1 | 0.27 | 0.51 | $A_1$ | 0.10 | 0.17 | 0.06 | 0.11 |
| $A_2$ | $\tilde{5}$ | $\tilde{1}$ | $\tilde{5}$ | $A_2$ | 6.58 | 1 | 6.58 | $A_2$ | 0.7 | 0.64 | 0.81 | 0.72 |
| $A_3$ | $\tilde{2}$ | $\tilde{5}^{-1}$ | $\tilde{1}$ | $A_3$ | 1.69 | 0.27 | 1 | $A_3$ | 0.18 | 0.17 | 0.12 | 0.16 |

**Table 14 The q-rung orthopair comparison matrix, normalized decision matrix and vector X w.r.t to C2.**

| q-rung orthopair decision matrix | $A_1$ | $A_2$ | $A_3$ | Decision matrix | $A_1$ | $A_2$ | $A_3$ | Normalized decision matrix | $A_1$ | $A_2$ | $A_3$ | Vector X |
|---|---|---|---|---|---|---|---|---|---|---|---|---|
| | | | | | | | | | | | | |
| $A_1$ | $\tilde{1}$ | $\tilde{1}$ | $\tilde{5}^{-1}$ | $A_1$ | 1 | 1 | 0.27 | $A_1$ | 0.11 | 0.44 | 0.02 | 0.19 |
| $A_2$ | $\tilde{1}$ | $\tilde{1}$ | $\tilde{6}$ | $A_2$ | 1 | 1 | 9.02 | $A_2$ | 0.11 | 0.44 | 0.87 | 0.47 |
| $A_3$ | $\tilde{5}$ | $\tilde{6}^{-1}$ | $\tilde{1}$ | $A_3$ | 6.58 | 0.26 | 1 | $A_3$ | 0.76 | 0.11 | 0.09 | 0.32 |

**Table 15 The q-rung orthopair comparison matrix, normalized decision matrix and vector X w.r.t to C3.**

| q-rung orthopair decision matrix | $A_1$ | $A_2$ | $A_3$ | Decision matrix | $A_1$ | $A_2$ | $A_3$ | Normalized decision matrix | $A_1$ | $A_2$ | $A_3$ | Vector X |
|---|---|---|---|---|---|---|---|---|---|---|---|---|
| | | | | | | | | | | | | |
| $A_1$ | $\tilde{1}$ | $\tilde{3}^{-1}$ | $\tilde{4}$ | $A_1$ | 1 | 0.24 | 4.6 | $A_1$ | 0.29 | 0.15 | 0.37 | 0.19 |
| $A_2$ | $\tilde{3}$ | $\tilde{1}$ | $\tilde{5}$ | $A_2$ | 2.05 | 1 | 6.58 | $A_2$ | 0.61 | 0.66 | 0.54 | 0.47 |
| $A_3$ | $\tilde{4}^{-1}$ | $\tilde{5}^{-1}$ | $\tilde{1}$ | $A_3$ | 0.3 | 0.27 | 1 | $A_3$ | 0.08 | 0.17 | 0.08 | 0.32 |

**Table 16 The q-rung orthopair comparison matrix, normalized decision matrix and vector X w.r.t to C4.**

| q-rung orthopair decision matrix | $A_1$ | $A_2$ | $A_3$ | Decision matrix | $A_1$ | $A_2$ | $A_3$ | Normalized decision matrix | $A_1$ | $A_2$ | $A_3$ | Vector X |
|---|---|---|---|---|---|---|---|---|---|---|---|---|
| | | | | | | | | | | | | |
| $A_1$ | $\tilde{1}$ | $\tilde{3}^{-1}$ | $\tilde{5}^{-1}$ | $A_1$ | 1 | 0.24 | 0.06 | $A_1$ | 0.10 | 0.10 | 0.02 | 0.19 |
| $A_2$ | $\tilde{3}$ | $\tilde{1}$ | $\tilde{1}$ | $A_2$ | 2.05 | 1 | 1 | $A_2$ | 0.21 | 0.44 | 0.48 | 0.47 |
| $A_3$ | $\tilde{5}$ | $\tilde{1}$ | $\tilde{1}$ | $A_3$ | 6.58 | 1 | 1 | $A_3$ | 0.68 | 0.44 | 0.48 | 0.32 |

**Table 17 The q-rung orthopair comparison matrix, normalized decision matrix and vector X w.r.t to C5.**

| q-rung orthopair decision matrix | $A_1$ | $A_2$ | $A_3$ | Decision matrix | $A_1$ | $A_2$ | $A_3$ | Normalized decision matrix | $A_1$ | $A_2$ | $A_3$ | Vector X |
|---|---|---|---|---|---|---|---|---|---|---|---|---|
| | | | | | | | | | | | | |
| $A_1$ | $\tilde{1}$ | $\tilde{2}^{-1}$ | $\tilde{3}$ | $A_1$ | 1 | 0.51 | 2.05 | $A_1$ | 0.34 | 0.28 | 0.26 | 0.29 |
| $A_2$ | $\tilde{2}$ | $\tilde{1}$ | $\tilde{4}$ | $A_2$ | 1.69 | 1 | 4.6 | $A_2$ | 0.57 | 0.55 | 0.60 | 0.57 |
| $A_3$ | $\tilde{3}^{-1}$ | $\tilde{4}^{-1}$ | $\tilde{1}$ | $A_3$ | 0.24 | 0.3 | 1 | $A_3$ | 0.08 | 0.16 | 0.13 | 0.12 |

Hence the final ranking of the best alternatives is $A_1 > A_3 > A_2$. The best alternative for the selection of best coffee brand is $A_1$. Alternatives along y-axis and ranking result along x-axis in Fig. 5.

**Table 18 The final ranking of alternatives.**

|        | $C_1$ | $C_2$ | $C_3$ | $C_4$ | $C_5$ | Total weight | Ranking |
|--------|-------|-------|-------|-------|-------|--------------|---------|
| Weight | 0.424 | 0.298 | 0.117 | 0.065 | 0.076 |              |         |
| $A_1$  | 0.71  | 0.47  | 0.60  | 0.37  | 0.57  | 0.57         | 1st     |
| $A_2$  | 0.11  | 0.19  | 0.27  | 0.07  | 0.29  | 0.16         | 3rd     |
| $A_3$  | 0.15  | 0.32  | 0.11  | 0.53  | 0.12  | 0.21         | 2nd     |

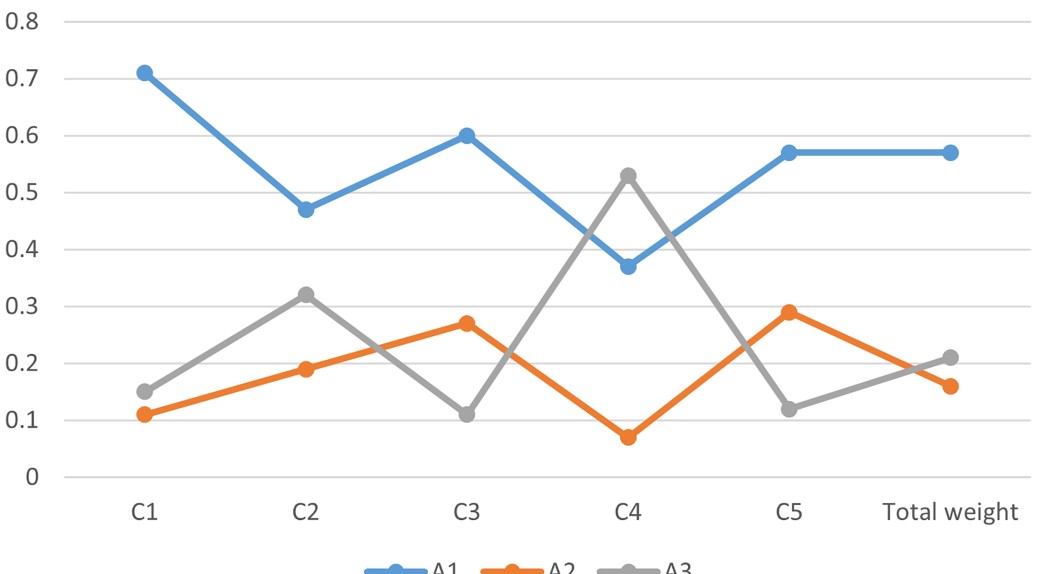

**Figure 5 Final ranking of alternatives using different criteria.**

**Example 3**: If we want to select the best coffee brands in South Africa, different factors affect this decision-making process, including cultural, social, and psychological. However, based on the following characteristics, like reliability, affordability, and quality, we can find the main factor affecting the decision of the best coffee brand. The hierarchical structure is presented in the figure given below. This hierarchical structure consists of three levels, wherein the first level, the objective, is selected. Criteria are offered at the second level, and alternatives are presented at the third level.

To construct hierarchy structure of the problem as in Fig. 3.

The q-rung orthopair comparison matrix with respect to criteria is given by:

$$\begin{bmatrix} . & C_1 & C_2 & C_3 \\ C_1 & 1^{\sim} & 2^{\sim} & 5^{\sim -1} \\ C_2 & 2^{\sim -1} & 1^{\sim} & 5^{\sim} \\ C_3 & 4^{\sim -1} & 4^{\sim} & 1^{\sim} \end{bmatrix}$$

The weight of criteria, sub-criteria, and alternatives as given below

**Table 19 q-rung orthopair comparison matrix.**

| $C_1$ | | $A_1$ | $A_2$ | $A_3$ | Priority vector | $C_2$ | | $A_1$ | $A_2$ | $A_3$ | Priority vector | $C_3$ | | $A_1$ | $A_2$ | $A_3$ | Priority vector |
|---|---|---|---|---|---|---|---|---|---|---|---|---|---|---|---|---|---|
| | $A_1$ | $\tilde{1}$ | $\tilde{5}^{-1}$ | $\tilde{2}^{-1}$ | 0.021 | | $A_1$ | $\tilde{1}$ | $\tilde{1}$ | $\tilde{5}^{-1}$ | 0.067 | | $A_1$ | $\tilde{1}$ | $\tilde{1}$ | $\tilde{5}^{-1}$ | 0.23 |
| | $A_2$ | $\tilde{5}$ | $\tilde{1}$ | $\tilde{5}$ | 0.88 | | $A_2$ | $\tilde{1}$ | $\tilde{1}$ | $\tilde{5}$ | 0.78 | | $A_2$ | $\tilde{1}$ | $\tilde{1}$ | $\tilde{6}$ | 0.65 |
| | $A_3$ | $\tilde{2}$ | $\tilde{5}^{-1}$ | $\tilde{1}$ | 0.045 | | $A_3$ | $\tilde{5}$ | $\tilde{6}^{-1}$ | $\tilde{1}$ | 0.43 | | $A_3$ | $\tilde{5}$ | $\tilde{6}^{-1}$ | $\tilde{1}$ | 0.96 |

**Table 20 The final ranking of the favorable alternative.**

| | $C_1$ | $C_2$ | $C_3$ | Global weight | Ranking |
|---|---|---|---|---|---|
| Weight | 0.087 | 0.655 | 0.258 | | |
| $A_1$ | 0.021 | 0.67 | 0.23 | 0.5 | 2nd |
| $A_2$ | 0.88 | 0.78 | 0.65 | 0.75 | 1st |
| $A_3$ | 0.045 | 0.43 | 0.96 | 0.34 | 3rd |

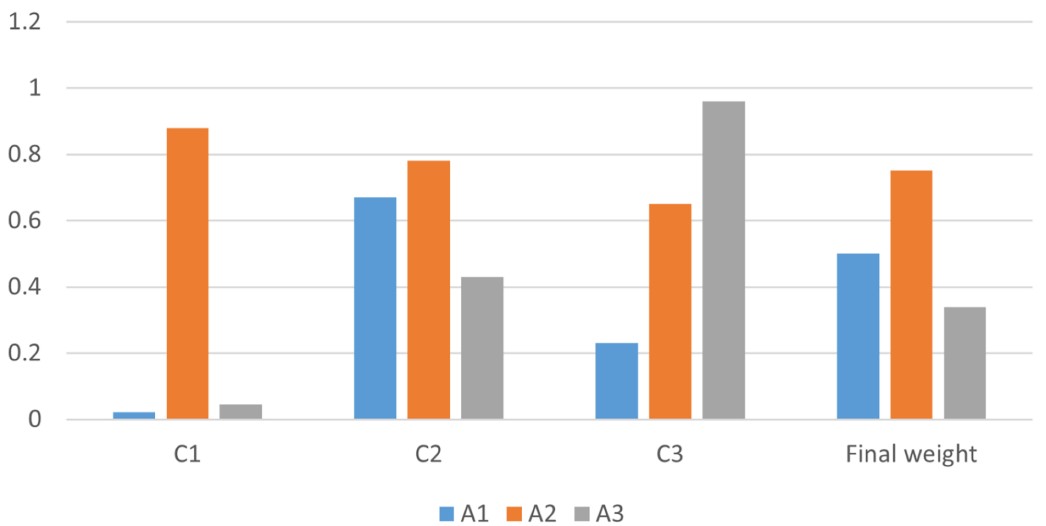

**Figure 6 Final ranking.**

$$\begin{bmatrix} C_1 & \begin{pmatrix} (0.0198, 0.1026, 0.3577), \\ (0.0314, 0.7299) \end{pmatrix} \\ C_2 & \begin{pmatrix} (0.1580, 0.6410, 2.8616), \\ (0.0314, 0.7299) \end{pmatrix} \\ C_3 & \begin{pmatrix} (0.0713, 0.2564, 0.7946), \\ (0.0404, 0.6526) \end{pmatrix} \end{bmatrix}$$

The weight vector and normalized weight vector is given respectively as:

$$\begin{bmatrix} 0.0482 \\ 0.3678 \\ 0.1450 \end{bmatrix}, \begin{bmatrix} 0.087 \\ 0.655 \\ 0.258 \end{bmatrix}$$

Now consider q-rung orthopair comparison matrix w.r. to $C_1$, $C_2$ and $C_3$ is given in the following table in Table 19.

The final ranking of the favorable alternative given below in Table 20.

Hence the final ranking of the alternatives is $A_2 \succ A_3 \succ A_1$. The final ranking shows that the social factor is the most influential factor affecting on the selection of best coffee brand while the cultural factor is the least influential. The graphical representation is shown in Fig. 6, where alternatives along y-axis and ranking result along x-axis in Fig. 6 as below.

## COMPARITIVE ANALYSIS AND CONCLUSION

### Comparitive analysis

The comparative analysis is conducted to check the validity of results. The above MCDM problem is evaluated by fuzzy AHP and method proposed in *Gulistan, Beg & Asif (2021)*. The findings are listed as under:

- Our approach yielded a final rank of $A_2$ with higher membership grades while the fuzzy AHP also yielded a final rank of $A_2$ with less membership values. Further the difference of ranking between $A_1$ and $A_3$ is higher in the case of fuzzy AHP while in q-rung orthopair fuzzy AHP the alternatives $A_1$ and $A_3$ are almost similar. This comparison highlights the superior effectiveness of our proposed method in this context.
- The method established in *Gulistan, Beg & Asif (2021)* yields the same ranking as obtained in this study.

The above comparison validate the results obtained by proposed study.

### Conclusion

Afica produces around 10% to 15% of world's total output. The African coffee market offers a rich and diverse range of coffee profiles. The coffee producers of Africa face numerous challenges like climate change, market fluctuations, diseases, soil degradation and limited access to finance. In this research, first we introduced the idea of q-rung ortho pair fuzzy AHP based on triangular q-rung fuzzy numbers. Secondly this newly proposed method is applied in identifying factors influencing consumers coffee brand preferences in South Africa. Complex decision-making is particularly challenging in scenarios with multiple layers and uncertain data. To test the applicability of the model we discuss the coffee brand selection specific to South Africa. The use of the developed model can provide more realistic selection of any alternative as it uses both membership and non-membership grades. The problem which we faced during its applicability is the justification of the non-membership grades in the real-life data. Secondly, we got reliable results, but we paid highly computational cost.

### Limitations

The decision making based on newly proposed method is computationally complex for a large scale problem. Therefore, there is a need of software programming to overcome the

computational cost. Furthermore the accuracy in results require expertise in fuzzy mathematics.

### Future directions

In future studies, we recommend expanding the sample size and brand selection and creating more control groups to investigate consumer behavior in coffee brand selection further. Further a comprehensive study is needed to reduce the computational cost. We are aiming to establish a PYTHON programming for computations. The hesitant fuzzy sets and fractional fuzzy sets are widely used for inconsistent data. In future, the AHP technique will be establish in fractional fuzzy sets and hesitant fuzzy sets.

### Funding

This research is funded by Researchers Supporting Project Number (RSPD2025R947), King Saud University, Riyadh, Saudi Arabia. The funders had no role in study design, data collection and analysis, decision to publish, or preparation of the manuscript.

### Grant Disclosures

The following grant information was disclosed by the authors:
Researchers Supporting Project Number: RSPD2025R947.
King Saud University, Riyadh, Saudi Arabia.

### Competing Interests

The authors declare that they have no competing interests.

### Author Contributions

- Yupei Huang conceived and designed the experiments, performed the computation work, prepared figures and/or tables, authored or reviewed drafts of the article, and approved the final draft.
- Muhammad Gulistan conceived and designed the experiments, performed the experiments, analyzed the data, performed the computation work, prepared figures and/or tables, authored or reviewed drafts of the article, and approved the final draft.
- Amir Rafique performed the experiments, analyzed the data, performed the computation work, prepared figures and/or tables, authored or reviewed drafts of the article, and approved the final draft.
- Wathek Chammam performed the computation work, prepared figures and/or tables, authored or reviewed drafts of the article, and approved the final draft.
- Khursheed Aurangzeb performed the computation work, prepared figures and/or tables, authored or reviewed drafts of the article, and approved the final draft.
- Ateeq Ur Rehman performed the computation work, prepared figures and/or tables, authored or reviewed drafts of the article, and approved the final draft.

## Data Availability

This is a methods article.

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
