# Peer review of "The technique of fuzzy analytic hierarchy process (FAHP) based on the triangular q-rung fuzzy numbers (TR-q-ROFNS) with applications in best African coffee brand selection"

_PeerJ Computer Science, doi:10.7717/peerj-cs.2555_

## Round 0.1 · original submission · Major Revisions

The author must improve the paper accordingly to all comments of the three reviewers.

Reviewer 1 ·

Basic reporting

poor. additional documents attached.

Experimental design

more in-depth analysis is required. additional documents attached.

Validity of the findings

no sensitivity analysis or comparisons with the other methods are included to validate the present methods outcomes. additional documents attached.

Additional comments

additional documents attached.

Annotated reviews are not available for download in order to protect the identity of reviewers who chose to remain anonymous.

Reviewer 2 ·

Basic reporting

The basic of this paper is very good.

Experimental design

The experiment results of this paper is clearly designed.

Validity of the findings

The validity of the finding is clear and compared with the existing results.

Additional comments

The authors of the paper describe their proposed approach for The Technique of Fuzzy AHP (FAHP) Based On The Triangular Q2 Rung Fuzzy Numbers (TR-Q-ROFNS) With Applications In Best Coffee Brand Selection. The topic is interesting and with possible applicability in the future. However, the paper needs several improvements:
1.An abstract should address these questions: what are you trying to do, why, what you found and what is the significance of your findings. Rewrite and improve.
2. There are several studies already developed in the literature. What is the utility of your proposed study is better than previous.
3. Comparative discussion needs further explanation. Please do more work on it.
4. Please improve the conclusion section. Also, limitations in the developed approach should be discussed in the conclusions section.
5. The notations used should be rechecked.
6. The writing is recommended to be improved. The authors are suggested to proofread paper and restructuring of sentences are required for the entire manuscript.

Reviewer 3 ·

Basic reporting

The paper is well-written and easy to follow. The introduction provides sufficient background on the topic and clearly states the research questions. The literature review is comprehensive and relevant. The methodology is described in detail, and the results are presented clearly. The conclusion summarizes the main findings and their implications.

Experimental design

no comment

Validity of the findings

no comment

Additional comments

I have a few suggestions to improve the manuscript:

In the introduction, the authors should provide more context on the South African coffee market. For example, they could discuss the size of the market, the major players, and the trends in coffee consumption.
In the literature review, the authors should discuss the limitations of previous studies on coffee brand selection. This would help to highlight the contribution of the current study.
In the methodology section, the authors should provide more information on the sampling method. For example, they could discuss how they selected the participants and how they ensured that the sample was representative of the South African population.
In the results section, the authors should provide more interpretation of the findings. For example, they could discuss the implications of the findings for coffee brands and marketers.
Compare the results with existing studies to highlight the study's contribution.
In the conclusion, the authors should discuss the limitations of the study and suggest directions for future research.
Suggest specific areas for future research to build on the findings of this study.

---

## Round 0.2 · accepted · Accept

The paper can be accepted.

Reviewer 1 ·

Basic reporting

no comment

Experimental design

it is ok now

Validity of the findings

everything is fine

Additional comments

no comments

Reviewer 2 ·

Basic reporting

Good

Experimental design

Good

Validity of the findings

Satisfied

Additional comments

The improved the paper according to my comments. Therefore, I accepted for publication.